# Study on Thermal Stability of Gel Foam Co-Stabilized by Hydrophilic Silica Nanoparticles and Surfactants

**DOI:** 10.3390/gels8020123

**Published:** 2022-02-15

**Authors:** Youjie Sheng, Yunchuan Peng, Shanwen Zhang, Ying Guo, Li Ma, Qiuhong Wang, Hanling Zhang

**Affiliations:** 1College of Safety Science and Engineering, Xi’an University of Science and Technology, Xi’an 710054, China; pengyunchuan33@163.com (Y.P.); 18615962035@163.com (S.Z.); wangqiuhong1025@126.com (Q.W.); zhanghanling920@163.com (H.Z.); 2College of Safety Science and Engineering, China University of Mining and Technology, Xuzhou 221116, China; guoying123@163.com; 3Yanzhou Coal Mining Co., Ltd., Zoucheng 237500, China

**Keywords:** nanoparticle, surfactant, gel foam, thermal stability, foam drainage, foam coarsening

## Abstract

The combination of nanoparticles (NP) and surfactant has been intensively studied to improve the thermal stability and optimize the performance of foams. This study focuses on the influence of silica NPs with different concentration on the thermal stability of gel foams based on a mixture of fluorocarbon (FS-50) and hydrocarbon (APG0810) surfactants. The surface activity, conductivity, viscosity, and foaming ability of the APG0810/FS-50/NPs dispersions are characterized. The effects of NP concentration on coarsening, drainage, and decay, as well as of the gel foams under thermal action, are systematically studied. Results show that NP concentration has a significant effect on the molecular interactions of the APG0810/FS-50/NP dispersions. The surface tension and conductivity of the dispersions decrease but the viscosity increases with the increase in NP concentration. The foaming ability of APG0810/FS-50 solution is reduced by the addition of NPs and decreases with the increase in NP concentration. The coarsening, drainage, and decay of the gel foams under thermal action slow down significantly with increasing NP concentration. The thermal stability of the gel foams increases with the addition of NPs and further increases with the increase in NP concentration. This study provides a theoretical guidance for the application for gel foams containing NPs and surfactants in fire-extinguishing agents.

## 1. Introduction

Aqueous foam is a disperse form of gas in a continuous liquid [1]. Foam is widely used in food [2], foam flotation [3], firefighting [4,5], and petroleum industries [6,7,8] because of its high fluidity, excellent rheology, low water consumption, and reduced formation damage [9,10,11]. Aqueous film-forming foam (AFFF) is currently recognized as the most widely used extinguishing agent in liquid fuel fires, as it has excellent foam stability and film-forming ability provided by fluorocarbon surfactants and hydrocarbon surfactants [12,13,14,15,16]. However, its core components, long-chain fluorocarbon surfactants (C8–C10), pose a serious threat to the ecological environment and are listed on the United Nations Environment Programme’s Controlled List of Persistent Organic Pollutants (POPs) [17,18]. Therefore, new efficient and environmentally friendly foam fire extinguishing agents have become the focus of international attention.

In the past few years, researchers have developed a series of environmentally friendly foam fire extinguishing agents by replacing long-chain fluorocarbon surfactants (C8–C10) with short-chain fluorocarbon surfactants (C4–C6) [19,20,21]. Hagenaars et al. analyzed the toxicity of short-chain fluorocarbons produced by DuPont and compared it with that of long-chain fluorocarbons. The toxicity of short-chain fluorocarbons was found to be much lower than that of long-chain fluorocarbons [13]. However, most of the environmentally friendly firefighting foams showed a decrease in stability under heat compared with the traditional AFFF on the market. Therefore, performance-enhanced foams resisting on high temperature needs to be studied.

Recently, nanoparticles have been applied to stabilize aqueous foams [22,23,24,25,26,27,28,29]. Both hydrophilic and hydrophobic NPs can stabilize the foam [30]. Researchers have conducted extensive studies on the stabilization mechanism of nanoparticle-stabilized foams [31,32,33,34]. Hydrophilic NPs slow down foam drainage and coarsening mainly by aggregating at the plateau borders, while hydrophobic NPs stabilize foam mainly by adsorbing at the gas-liquid interface of the foam film [35,36]. However, most of the current studies have just focused on the foam stability of NPs with a single surfactant. Yet, in practice, because of the synergistic effect between different types of surfactants, the mixed surfactant system is usually more advantageous than the single surfactant system [37,38,39]. Besides, few study focuses on nanoparticle-stabilized foams under high temperature. Therefore, considerable works need to be conducted to investigate the effect of NPs on foam thermal stability of multi-component surfactants.

In this work, the gel foams consisting of an amphoteric fluorocarbon surfactant, nonionic hydrocarbon surfactants, and silica NPs with different concentration are prepared. The foam drainage and foam decay of the mixed dispersions at high temperatures are systematically studied. The effect of different concentrations of NPs on foam coarsening of the mixed dispersions at different temperatures is analyzed deeply. The results of this work provide a theoretical foundation for the application of NPs in firefighting foams.

## 2. Results and Discussion

Table 1 shows the formulations of the gel foam dispersions, consisting of nonionic hydrocarbon surfactants (APG0810), amphoteric fluorocarbon surfactant (FS-50), NaCl, NPs, and deionized water.

### 2.1. Basic Properties of Gel Dispersions

Figure 1 shows the equilibrium surface tension, electrical conductivity, and dynamic viscosity of the five gel dispersions. It should be pointed out that the surface tension of A-4# cannot be obtained by the method used in the present study due to its high viscosity and poor flowability. The surface tensions of A-1#, A-2#, and A-3# are around 18.4 mN/m, obviously higher than that of A-0#. The results show that the addition of NPs significantly reduces the surface activity of the FS-50/APG0810 solutions. This should be ascribed to the decrease in the number of surfactant molecules at gas/liquid interface due to the addition of NPs. Previous studies indicated that the surfactant molecules tended to adsorb onto the surface of SiO_2_ NPs through silanization reactions, electrostatic interactions, and hydrogen bonding [40,41]. The adsorption equilibrium of surfactant molecules at the gas/liquid interface was disrupted by the addition of NPs [42,43,44,45]. The surfactant molecules are transferred onto the surface of NPs from gas/liquid interface, resulting in decrease in surface activity. It is worth noting that the concentration of NPs has little effect on the surface activity of the mixed dispersion. This is mainly due to the formation of hydrophobic aggregates caused by adsorption of hydrophilic groups of surfactant molecules on the NP surface. The hydrophobic groups of surfactant molecules are far away from the NP surface, leading to the formation of hydrophobic aggregates containing NPs and surfactants. The hydrophobic aggregates tend to go back to gas/liquid interface, resulting the slight increase in surface activity.

The conductivity of the mixed dispersions decreases with increasing NP concentration, and the viscosity increases with increasing NP concentration. The decrease in conductivity further verifies that part of the surfactant is adsorbed onto the surface of the NPs, resulting in a reduction of the free charge in solution. The increase in viscosity of the mixed dispersions is due to the addition of a large number of NPs and the molecular interaction between the NPs and the surfactant [46].

The initial foaming heights of the five gel dispersions are shown in Figure 2. The foaming ability of the mixed dispersions decreases with the increase in NP concentration. Notably, A-4# is too viscous and has very poor flowability, so is unable to generate foam. Many factors have an influence on foaming ability, including surface activity, viscosity, molecule structure of surfactants, electrostatic charges, and so on. Increase in viscosity and decrease in surface activity prevented the formation of bubbles [47,48]. In the present study, the reduction of foaming ability is mainly due to the rapid increase in viscosity and decrease in surface activity.

### 2.2. Foam Thermal Stability

#### 2.2.1. Foam Drainage and Decay

Figure 3 presents the typical process of foam decay under thermal radiation. The foam decay of A-0# and A-2# shows the same trend; that is, the foam height decreases following a short increase. Gas expansion in foams under heat results in the short increase in foam volume. Compared to A-2#, A-0# shows the faster and longer foam expansion. The A-0# foam expands to the upper surface of the container at 5 min, while the corresponding time of A-2# is 10 min. Notably, the foam decay rate of A-2# is slower than that of A-0#. At 60 min, the foam height of A-0# decreases by half but that of A-2# remains more than two-thirds. These results indicate that the addition of NPs obviously slow down the foam stability under heat.

In order to quantitatively analyze foam stability, the variation in foam height and drainage height of the dispersion under thermal radiation versus time is plotted in Figure 4. Overall, the foam drainage curves of the five groups of gel dispersions rise and then flatten gradually. The foam drainage curves of the mixed dispersions descend rapidly with the increase in NP concentration. Among them, the drainage curves of A-4# have extremely small variation and are much lower than the other four groups of curves. In contrast to the foam drainage curves, the foam height curves decrease following a short increase over time. The short increase is corresponding the foam expansion during foam thermal stability experiment. The foam height curves rise gradually with increasing NP concentration. The foam decay curves of A-1#, A-2# and A-3# are relatively close. The foam decay curve of A-0# is much lower than other curves. The foam height curve of A-4# is much higher than other four groups. After 60 min, the foam height of A-4# is higher than its initial value. This is because the foam of A-4# decays and drains out liquid very slowly after foam expansion. The increase in foam volume due to gas expansion under heat has not disappeared after 60 min yet. The result demonstrates that the gel foam containing NPs with a high concentration is highly stable under thermal radiation.

In order to further analyze the heat insulation ability of the gel foams, the temperatures obtained from thermocouples at different depths of foams are collected and plotted versus time in Figure 5. In general, the higher the concentration of NPs in the foam dispersions, the smaller the temperature obtained at the same depth at the same moment. The temperature curves of K-1, K-2, and K-3 show the same trend, keeping a relatively stable after a rapid increase. The temperature curves of K-1 are relatively densely distributed. For K-2 and K-3, the temperature curves of A-2#, A-3#, and A-4# are closer and significantly lower than those of A-0# and A-1#. In addition, the temperature curves of A-0# and A-1# in K-3 are closer than that of K-2. For the depth of K-4, the temperature curves of all the five gel foams reach a relatively stable value after a rapid increase. Then, the temperature curves increase sharply again and then reach the second relatively stable value. The second rapid increase in temperature curves of K-4 is caused by the complete collapse of foam layer around thermocouple. The thermocouples of K-4 are exposed to the air once the foam layer around K-4 collapse completely, resulting in the second rapid increase. In the curves of K-4, the complete foam collapse times of A-0#, A-1#, A-2#, and A-3# are denoted as t0, t1, t2, and t3 respectively. With the increase in NP concentration, the foam collapse time gradually prolongs. Notably, no second rapid increase occurs in temperature curve of A-4# at K-4, indicating that the K-4 thermocouple of A-4# is still buried in the foam layer during the whole experiment.

#### 2.2.2. Foam Coarsening

The typical foam coarsening process under different temperature is shown in Figure 6. The gel foams have a close initial bubble size. The bubble size becomes larger and the number of bubbles decreases over time due the Laplace pressure [49]. With time goes on, the foam coarsening processes of the three gel dispersions at different temperatures show different morphological changes. At 45 °C, the bubbles of A-0# have been changed from an elliptical three-dimensional structure to a polygonal two-dimensional structure at 6 min, and the bubbles largely disappears at 8 min. The bubbles of A-1# are transformed into a two-dimensional structure at 8 min and the bubble starts to break at 16 min. The bubble of A-3# still maintains a relatively regular two-dimensional structure at 8 min. Both A-1# and A-3# have a certain amount of water left after the foam burst, as indicated by the arrows in Figure 7. The foam coarsening process at 85 °C is similar to that at 45 °C. The higher temperature results in the faster increase in bubble size and disappearance. These results indicate that the addition of NPs significantly delays the foam coarsening under heat. In addition, the temperature has a significant effect on the foam coarsening of the mixed gel dispersions.

The normal distributions of bubble diameter relative to frequency quantify the difference in bubble size under different conditions, and are able to quantitatively analyze the foam-coarsening process [50]. Figure 7 shows the variation in foam size distribution versus time during foam coarsening at different temperatures for A-0#, A-1#, and A-3#. At 1 min, the foam size distributions of the three mixed dispersions are similar. The normal distribution curves of the bubbles are limited in 100 µm at the moment, implying that the initial diameter of bubbles is distributed in 100 µm. This result is consistent with the previous findings [35]. The bubble size distributions gradually became wider as time progressed. At the same temperature and with different samples, the foam size distribution of A-0# widens faster, and the foam size curve is closer than that of A-1# and A-3#. Foam size distribution of the dispersion widens faster and the number of the foam size curves increase as the temperature increases. Moreover, the foam lasting time during coarsening process is reduced. This is because the higher the temperature is, the faster the evaporation of water vapor, and the corresponding bubbles quickly become thicker, larger, and quickly broken. The results show that NPs have a significant deceleration on the foam-coarsening rate under heat. The higher the concentration of NPs in gel foams is, the stronger the deceleration effect.

In order to quantify the bubble coarsening process, the dimensionless average area ($\bar{A}$) of individual bubbles at different temperatures is measured and plotted versus time in Figure 8. In general, the $\bar{A}$ of the gel dispersions increases over time. The higher concentration of NPs leads to the slower change of the $\bar{A}$. In addition, the foam lasting times of all the five gel dispersions gradually decrease with the increase in temperature. At 25 °C, the $\bar{A}$ curves almost overlap each other. At 45 °C and 65 °C, the $\bar{A}$ curves show an obviously change. The $\bar{A}$ of A-0#, A-1# and A-2# under 45 °C has the maximum value among the four temperature experiments. A-3# and A-4# reach the maximum at 65 °C. This is mainly because the bubbles have disappeared completely before reaching a bigger $\bar{A}$ due to the higher temperature of 65 °C and 85 °C. The results show that the presence of NPs significantly slows down the bubble growth under heat. In addition, this effect is enhanced with the increase in NP concentration.

The foam stabilization time (maximum value of foam lasting time during coarsening process) of the gel dispersions at different temperatures is shown in Figure 9. The foam stabilization time of the gel dispersion increases with the increase in NP concentration. In addition, the foam stabilization time curves show a linear increase over NP concentration. Hence, the data of the foam stabilization time are linearly fitted. The fitting lines are in good agreement with the original curves. The fitting results are as follows:

25 °C: y=4.24x+31.44; 45 °C: y=3.38x+10.39;

65 °C: y=2.27x+4.12; 85 °C: y=1.11x+2.38.

**Figure 9 gels-08-00123-f009:**
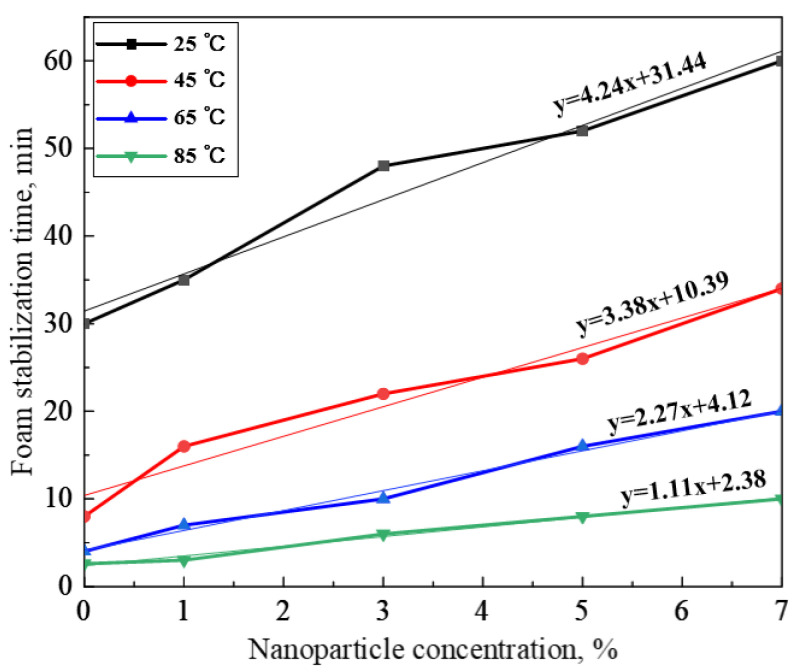
Foam stabilization time of dispersions at different temperatures.

The slope of the fitting equations under different temperature reflects the change rate of foam coarsening versus NP concentration. The bigger value means a stronger effect of NP concentration on foam coarsening. The foam stabilization time curves of the gel dispersion drop with elevated temperature. The slope decreases with elevated temperature, indicating that the deceleration effect of NPs on foam coarsening decrease under high temperature. The result should be ascribed to the change of foam coarsening process under high temperature. At a high temperature (such as 85 °C), foam coarsening is not only driven by Laplace pressure but also by bubble film breaking under high temperature. Bubbles break before becoming bigger bubbles under high temperature.

Obviously, the addition of NPs can significantly improve the thermal stability of the gel foams co-stabilized by NPs and surfactants. Many mechanisms have been proposed to account for the stability of foams stabilized by a mixture of surfactants and NPs at room temperatures [31,32,33,34,51]. It is generally accepted that NPs interact with surfactants and aggregate in large numbers at liquid films and plateau borders. The complicated aggregates containing NPs and surfactants increased the maximum capillary pressure, and thus slowed down the bubble drainage and coarsening [4,33,51,52]. The stability mechanism of the gel foams under thermal action is more complex than that at room temperature. Foams collapse at high temperature due to drainage, coarsening, and evaporation. The added NPs improve the thermal stability of the foam mainly by slowing down the foam drainage and foam coarsening. The slower rate of foam drainage allows more water to be retained in the foam layer. The retained water prevented the heat transfer through the foam layer. In addition, the added NPs accumulate in the liquid film and plateau borders, forming a solid network structure around the bubble, contributing to thermal insulation [30,53]. At the same time, the presence of the NP solid network structure enhances the ability of the foam liquid film to resist external interference. These factors commonly promote the enhancement of the gel foam thermal stability.

## 3. Conclusions

The thermal stability of the foams stabilized by NPs, FS-50, and APG0810 were investigated systematically. The NPs have a significant influence on drainage, decay, and coarsening of the gel foams. The main conclusions are as follows:

Intense molecular interactions exist between surfactant molecules and NPs. The concentration of NPs has a significant effect on molecular interactions. Surface tension and electrical conductivity of the gel dispersions decrease with increasing NP concentration, and viscosity increases with increasing NP concentration. The foaming ability of APG0810/FS-50 solution decreases with addition of NPs and further decreased with the increase in NPs concentration.

The presence of NPs greatly enhances the foam thermal stability of the mixed APG0810/FS-50 solution. The thermal stability of the gel foams containing APG0810/FS-50/NPs further increases with the increase in NP concentration. The foam stability of APG0810/FS-50/NPs dispersions decreased rapidly with elevated temperature. The effect of NP concentration on the foam stability of dispersions decreases with elevated temperature.

The added NPs improve the thermal stability of the foam mainly by aggregating and forming network structures with surfactant molecules in the liquid film and plateau borders of the bubbles. The formed network structures slow down the foam drainage, foam coarsening, and, hence, hinder the heat transfer in the foam layer and enhance the ability of the foam liquid film to resist external disturbances.

## 4. Materials and Methods

### 4.1. Sample Preparation

The mixed dispersions are composed of APG0810 (Shandong Yousuo Chemical Technology Co., Ltd, Shandong, China), FS-50 (DuPont, Guangzhou, China), NaCl, hydrophilic gas-phase silica NPs (Shanghai Aladdin Bio-Chem Technology Co., Ltd, Shanghai, China), and water (Table 1). The molecular formulas of FS-50 and APG0810 are shown in Figure 10. APG0810 and FS-50 are present in the mixed dispersions at concentrations exceeding their own critical micelle concentrations of 2 mM [54] and 0.0126% [55], respectively. The APG0810, FS-50, NaCl, and NPs are added into the container sequentially under magnetic stirring for 10 min between each addition. The mixtures are put into the Scientz-2400F ultrasonic disperser and dispersed for 10 min to obtain the homogeneous foam dispersions. The mixed dispersions are numbered as A-0#, A-1#, A-2#, A-3#, and A-4# according to the NP concentration.

### 4.2. Apparatus and Procedures

#### 4.2.1. Testing for Dispersions Properties

The dynamic viscosity of the mixed dispersions is measured by DV-1 digital viscometer (Shanghai Fangrui Instrument Co., Ltd., Shanghai, China). The electrical conductivity of the mixed dispersions is measured by a SG23-B Mettler multiparameter tester (Mettler Toledo, Shanghai, China). The surface tension is measured using a QBZY-3 fully automatic surface tension meter (Shanghai Fangrui Instrument Co., Ltd., Shanghai, China) based on the platinum plate method in a water bath. During testing, the platinum plate is moved towards the upper surface of dispersions from the top. The platinum plate stops moving once it touches the upper surface of dispersions. The surface tension increases gradually from this point. After a period of time, the surface tension become stable gradually, and the corresponding stable value is the equilibrium surface tension. All the tests are repeated three times to obtain an average.

#### 4.2.2. Testing for Foaming Ability

Foaming ability is an important index of foam fire extinguishing agent, which has an important influence on the fire extinguishing efficiency of foam fire extinguishing agent. Ross-Miles is a standard method for evaluating foaming performance [56]. The improved Ross-Miles method is used to measure the foaming ability of gel dispersions in the present study. The apparatus diagram and testing procedure in details can be seen in the reference [57]. The 200 mL of dispersion is added to the top spherical funnel device, and the dispersion falls through a height of 1 m into a cylindrical glass container containing 50 mL of the dispersion to complete the foaming process. The initial foaming height created by the gel dispersions is recorded to characterize the foaming ability of the dispersion.

#### 4.2.3. Testing for Foam Thermal Stability

##### Testing for Foam Drainage and Foam Decay

The device comprises a heat source, a CCD camera, a foam container (L × W × H = 3 cm × 2 cm × 15 cm), a data collector, and a thermocouple tree. The radiant heat source used in the experiment is provided by a SGM series radiant heat source with a rated power of 600 W, purchased from Luoyang Saina Thermal High Temperature Technology Co., Ltd (Luoyang, China). The apparatus diagram and detailed test procedure can be found in ref. [58]. Four K-type armored thermocouples, numbered as K-1, K-2, K-3, and K-4, are evenly distributed in the foam container with a distance of 3 cm from the bottom to the top of the foam container. During testing, 80 mL foam is injected into the foam container. Temperature data is collected by a computer. The whole process is recorded by a CCD camera, and the video data is processed by Image J software to obtain the instantaneous data of foam volume decay and the foam drainage height.

##### Testing for Foam Coarsening

The foam coarsening experiments under thermal action are performed on a transparent quartz heating plate (L × W = 7 cm × 5 cm) with a model of TY83-SC. The experimental temperature can be changed from 25 °C to 200 °C. In the present study, the temperature is controlled at 25 °C, 45 °C, 65 °C, and 85 °C, respectively. The 2 mL foams are added into the center of the Petri dish and then placed on the quartz heating plate. One image is taken per minute after covering a slide.

## Figures and Tables

**Figure 1 gels-08-00123-f001:**
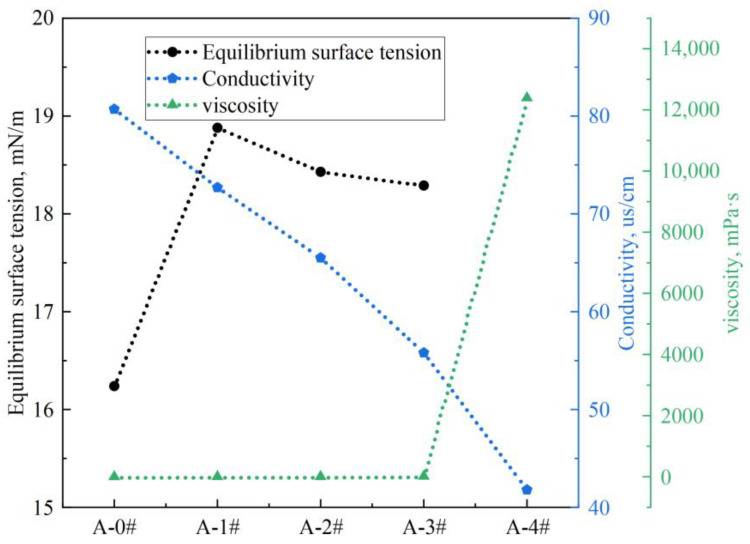
Equilibrium surface tension, electrical conductivity, and viscosity of five foam dispersions.

**Figure 2 gels-08-00123-f002:**
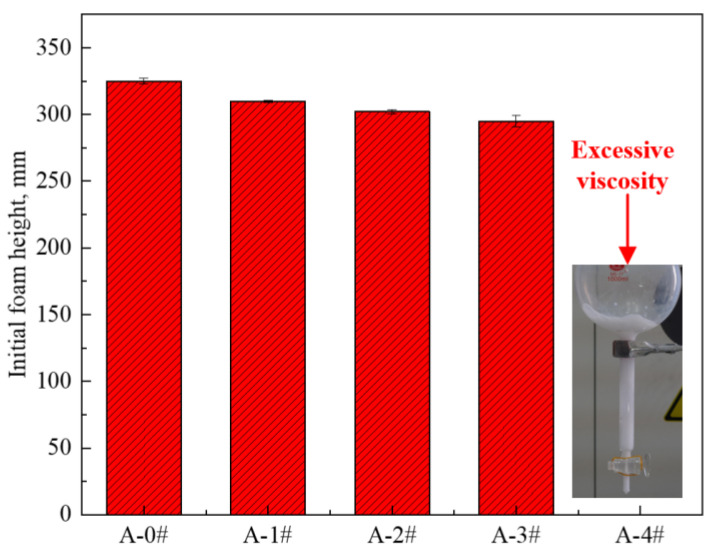
Initial foam height of five foam dispersions.

**Figure 3 gels-08-00123-f003:**
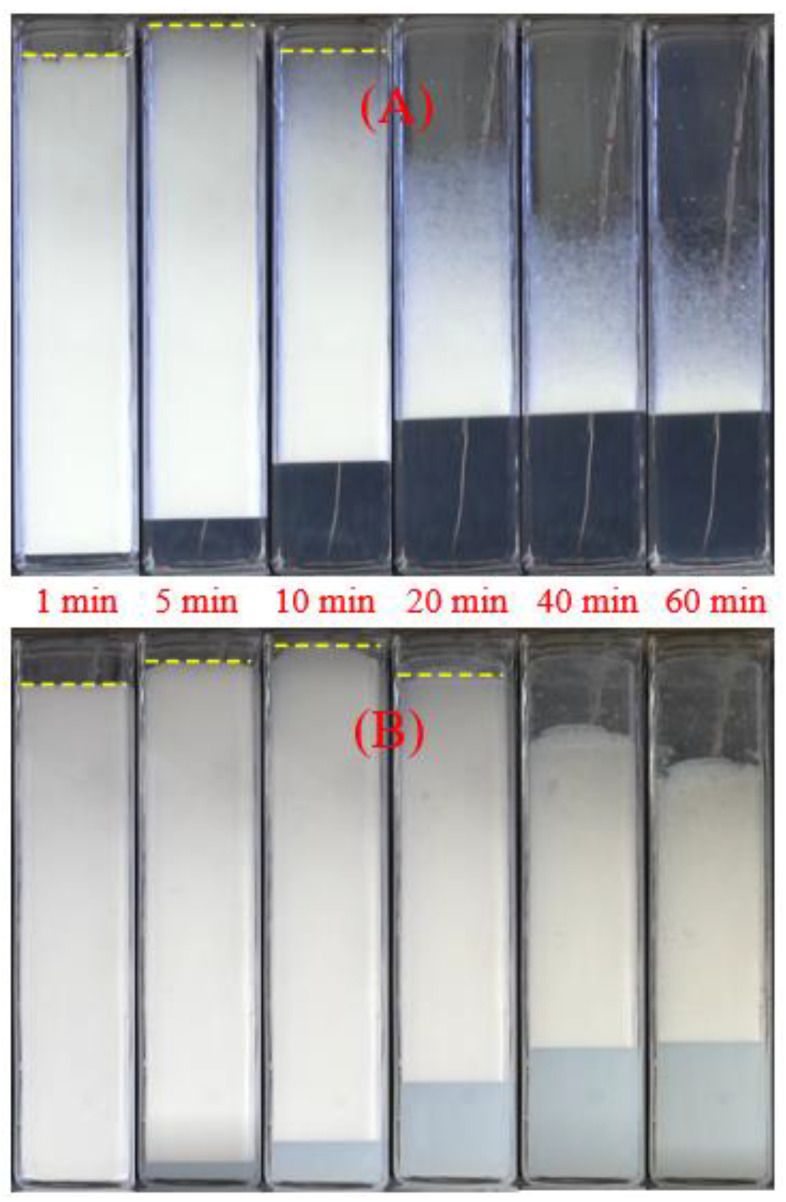
The appearance change of the foams exposed to thermal radiation: (**A**) A-0#; (**B**) A-2#.

**Figure 4 gels-08-00123-f004:**
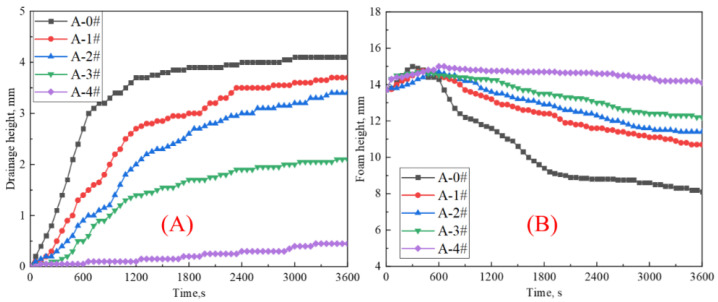
(**A**) The height of drainage varies with heating time. (**B**) Foam layer height varies with heating time.

**Figure 5 gels-08-00123-f005:**
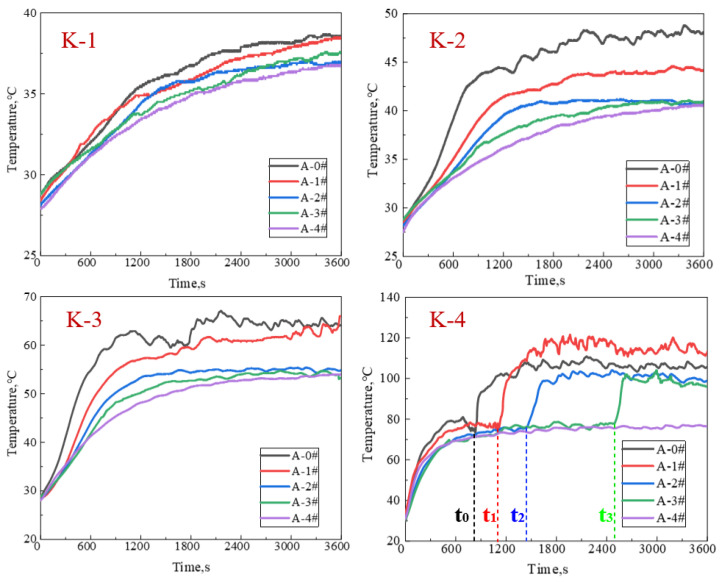
Temperature distribution inside foam layer at different positions under 200 °C: the distance of K-1, K-2, K-3, and K-4 from the heat source are 12, 9, 6, and 3 cm, respectively.

**Figure 6 gels-08-00123-f006:**
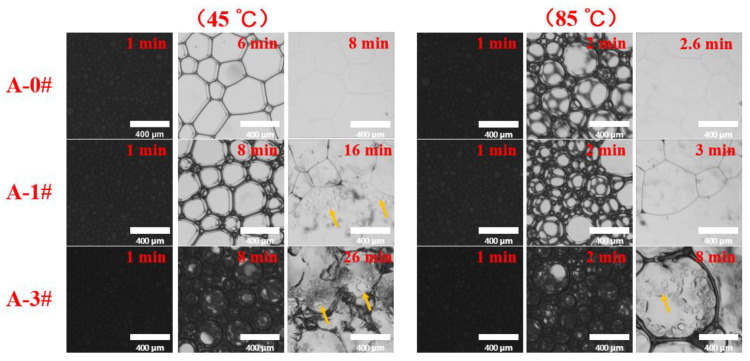
Morphology of bubbles during foam coarsening under thermal action.

**Figure 7 gels-08-00123-f007:**
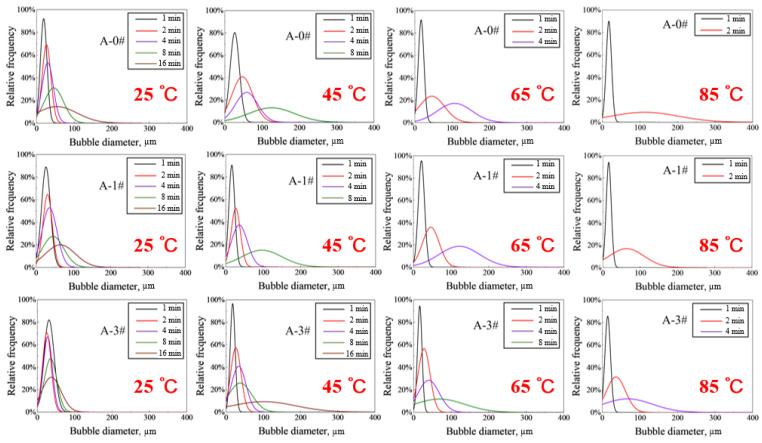
Relative frequency of bubble size distribution of A-0#, A-1#, and A-3#.

**Figure 8 gels-08-00123-f008:**
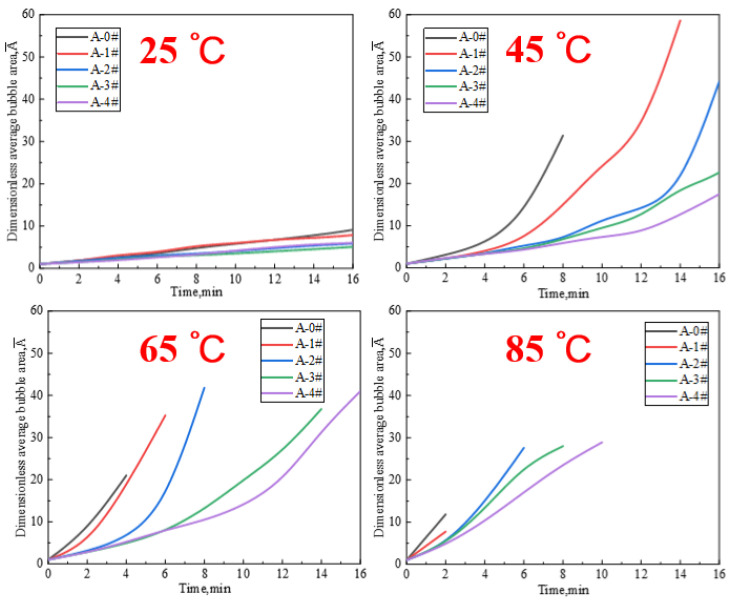
Variation of uncaused mean foam numbers in mixed dispersions at different temperatures with time.

**Figure 10 gels-08-00123-f010:**
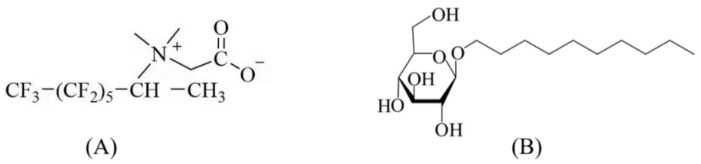
Molecular structures of: (**A**) FS-50 and (**B**) APG0810.

**Table 1 gels-08-00123-t001:** Foam dispersions with various compositions.

Samples	APG0810 (%)	FS-50 (%)	NaCl (mM)	NPs (%)
A-0#	0.5	0.25	1	0
A-1#	0.5	0.25	1	1
A-2#	0.5	0.25	1	3
A-3#	0.5	0.25	1	5
A-4#	0.5	0.25	1	7

## Data Availability

Data are contained within the article.

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
