# Peer review of "Study on Thermal Stability of Gel Foam Co-Stabilized by Hydrophilic Silica Nanoparticles and Surfactants"

_gels, 2022, doi:10.3390/gels8020123_

Round 1

Reviewer 1 Report

The paper under review is devoted to study of improving the foam stability by using the synergetic effect of nanoparticles and surfactants.

This is a quite reasonable approach having both theoretic and applied value.

The work was carried out accurately and contains all necessary components including a comprehensive literature review, detail description of experimental results, and discussion of the obtained data.

I hve only one comment. Presentation of the experimental data in Fig. 8 by broken lines connecting points is not a food method, especially without giving the confidence limits of the points. I would prefer to see smooth averaging curves.

Author Response

Response to Reviewer 1 Comments

Comments: The paper under review is devoted to study of improving the foam stability by using the synergetic effect of nanoparticles and surfactants. This is a quite reasonable approach having both theoretic and applied value. The work was carried out accurately and contains all necessary components including a comprehensive literature review, detail description of experimental results, and discussion of the obtained data.

Response:

Thank you for your kind comments concerning our manuscript. We would like to extend our hearty thanks for your excellent work on our paper. The revisions were addressed point by point as below.

Point 1: I have only one comment. Presentation of the experimental data in Fig. 8 by broken lines connecting points is not a food method, especially without giving the confidence limits of the points. I would prefer to see smooth averaging curves.

Response:

Thank you very much for your kind review. We have modified the line style in Figure 8 following your advice. Please check it in the revised manuscript.

Reviewer 2 Report

The present manuscript entitled “Study on thermal stability of gel foam co-stabilized by hydrophilic silica nanoparticles and surfactants” by Youjie Sheng et al., describe the influence of silica NPs with a different concentration on the thermal stability of gel foams based on a mixture of fluorocarbon (FS-50) and hydrocarbon (APG0810) surfactants. Furthermore, Surface activity, conductivity, viscosity, foaming ability of the APG0810/FS-50/NPs dispersions were characterized. This present study delivers theoretical guidance for the application of gel foams containing NPs and surfactants in fire-extinguishing agents. The authors report an interesting approach. The objective and justification of the work are clear, and the experimental work is significant. The study is very accurate and adequate, and thus, I recommend it for publication. However, certain Minor issues are detailed below which need to be addressed before its final acceptance in gels.

Comment 1:  There are so many typographical errors in the manuscript text, authors must check typos, use of is, are, was, were, and preposition.

Comment 2: The abstract is poorly written, should be edited. It must summarize well the obtained results.

Comment 3: In the introduction section add some more recent literature to strengthen the section.

Comment 4: Include the Graphical Abstract for the manuscript.

Comment 5: Figure 5 caption should be changed as shown details in the figure, include the details of  K-1,  K-2, K-3, and K-4 in the figure caption.

Comment 6: Relative frequency of bubble size distribution of the A-0#, A-1#, and A-3# (Figure 7) results should be discussed wider with some more references in order to get better results discussion.

Comment 7: The conclusions section is too short, the authors should revise it.

Author Response

Response to Reviewer 1 Comments

Comments: The present manuscript entitled “Study on thermal stability of gel foam co-stabilized by hydrophilic silica nanoparticles and surfactants” by Youjie Sheng et al., describe the influence of silica NPs with a different concentration on the thermal stability of gel foams based on a mixture of fluorocarbon (FS-50) and hydrocarbon (APG0810) surfactants. Furthermore, Surface activity, conductivity, viscosity, foaming ability of the APG0810/FS-50/NPs dispersions were characterized. This present study delivers theoretical guidance for the application of gel foams containing NPs and surfactants in fire-extinguishing agents. The authors report an interesting approach. The objective and justification of the work are clear, and the experimental work is significant. The study is very accurate and adequate, and thus, I recommend it for publication. However, certain Minor issues are detailed below which need to be addressed before its final acceptance in gels.

Response:

Thank you for your insightful comments concerning our manuscript. We would like to extend our hearty thanks for your excellent work on our paper. The revisions were addressed point by point as below.

Comment 1: There are so many typographical errors in the manuscript text, authors must check typos, use of is, are, was, were, and preposition.

Response:

Thank you very much for your careful and instructive advice. We have carefully checked and modified the errors according to your suggestions. Please check them.

Comment 2: The abstract is poorly written, should be edited. It must summarize well the obtained results.

Response:

Thank you very much for your valuable suggestions. We have carefully revised the abstract following your suggestions.

Comment 3: In the introduction section add some more recent literature to strengthen the section.

Response:

Thank you very much for your careful and instructive advice. We have added some recent literature in the introduction section in revised manuscript. The newly added references are [7], [8], [15], [21], [28], [29], and [36] in the revised manuscript. Please check it.

Comment 4: Include the Graphical Abstract for the manuscript.

Response:

Thank you very much for remind. We have sent a graphical abstract to editor when we submitted the manuscript online.

Comment 5: Figure 5 caption should be changed as shown details in the figure, include the details of K-1, K-2, K-3, and K-4 in the figure caption.

Response:

Thank you very much for your careful and instructive advice. We have modified the caption of Figure 5 based on your suggestions, and added information about K-1, K-2, K-3 and K-4 thermocouples to the revised caption. The new title is as follows: “Temperature distribution inside foam layer at different positions under 200℃: the distance of K-1, K-2, K-3, and K-4 from the heat source are 12, 9, 6, and 3 cm, respectively.”.

Comment 6: Relative frequency of bubble size distribution of the A-0#, A-1#, and A-3# (Figure 7) results should be discussed wider with some more references in order to get better results discussion.

Response:

Thank you for your valuable suggestions. We have discussed the data in Figure 7 more extensively based on your comments, describing the prevalent phenomenon at the initial moment of the bubble and explaining the reasons for faster foam breakage at high temperatures. And we have added previous studies as an effective support for our research results. Please check it.

Comment 7: The conclusions section is too short, the authors should revise it.

Response:

Thank you very much for your valuable suggestions. We have carefully revised the conclusion section in the revised manuscript.